# Virtual Commissioning of Distributed Systems in the Industrial Internet of Things

**DOI:** 10.3390/s23073545

**Published:** 2023-03-28

**Authors:** Julia Rosenberger, Andreas Selig, Mirjana Ristic, Michael Bühren, Dieter Schramm

**Affiliations:** 1Bosch Rexroth AG, 97816 Lohr am Main, Germany; 2Institute for Mechatronics Bocholt, Westfälische Hochschule, 46395 Bocholt, Germany; 3Faculty of Engineering, University of Duisburg-Essen, 47057 Duisburg, Germany

**Keywords:** Industrial Internet of Things (IIoT), simulation, hardware-in-the-loop (HiL), virtual commissioning (VICO), distributed stream processing system (DSPS), edge computing

## Abstract

With the convergence of information technology (IT) and operational technology (OT) in Industry 4.0, edge computing is increasingly relevant in the context of the Industrial Internet of Things (IIoT). While the use of simulation is already the state of the art in almost every engineering discipline, e.g., dynamic systems, plant engineering, and logistics, it is less common for edge computing. This work discusses different use cases concerning edge computing in IIoT that can profit from the use of OT simulation methods. In addition to enabling machine learning, the focus of this work is on the virtual commissioning of data stream processing systems. To evaluate the proposed approach, an exemplary application of the middleware layer, i.e., a multi-agent reinforcement learning system for intelligent edge resource allocation, is combined with a physical simulation model of an industrial plant. It confirms the feasibility of the proposed use of simulation for virtual commissioning of an industrial edge computing system using Hardware-in-the-Loop. In summary, edge computing in IIoT is highlighted as a new application area for existing simulation methods from the OT perspective. The benefits in IIoT are exemplified by various use cases for the logic or middleware layer using physical simulation of the target environment. The relevance for real-life IIoT systems is confirmed by an experimental evaluation, and limitations are pointed out.

## 1. Introduction

The current trend of Industry 4.0 (I4.0) and Industrial Internet of Things (IIoT) has caused a high number of data-based business models and digital services such as cloud computing or predictive maintenance. One principle of I4.0 is distributed decision making [1] that is enabled by the IIoT and data stream systems. While cloud computing is still the predominantly used method for data processing, in many cases in industry, it is not sufficient in terms of latency, Quality of Service (QoS), availability, and reliability. In addition, many end-users still object to transferring critical enterprise and production data to remote service providers. Thus, the shift to edge computing has been recognizable for a few years in industry [2,3,4,5]. In the context of distributed stream processing systems (DSPSs), this trend is also evident, as the use of edge device resources for (pre)processing tasks is increasing [6,7]. The new requirements also entail an effect in the development of industrial edge devices, as connectivity, new programming languages, and open standards are increasingly supported [8]. For control units, in [9], the progress regarding the capabilities to run machine learning algorithms is described. Nevertheless, one of the most limiting factors is the resource constraints of the edge devices. In the vision of I4.0 and IIoT, devices are fully connected, and software is run in distributed systems. The convergence of Information Technology (IT) and Operational Technology (OT) is highlighted several times in the context of I4.0. However, this is mostly related to new data-driven applications and business models.

The design and implementation of distributed interacting computing systems is of high complexity. Thus, for DSPSs, the authors in [10] already highlighted the lack of performance evaluation of DSPSs under realistic simulation of the processing workload and environment. With regard to the challenges of limited edge computing resources, as well as limited communication resources and the possibility of dynamic changes in the network, a simulation of the target system for early validation is of great benefit.

Our motivation for this study arose from a situation where the testing of a new generation DSPS applying edge devices and agent-based task distribution was required, but the real-world environment was not yet available. A satisfactory solution could be achieved through the realistic simulation of an exemplary industrial plant that exchanges data streams via Hardware-in-the-Loop (HiL). This realization has motivated us to highlight the broad application field for existing OT simulations to advance the development of edge computing systems in IIoT. This paper is intended to encourage the consideration of the convergence of the two worlds, IT and OT, in validation and verification methods. While many of the emerging applications are still validated with the state-of-the-art methods of the IT world, e.g., network simulators, less attention is paid to the existing tools of the OT world. For some systems, especially edge-computing-based ones, there is a lack of simulation options, as they also require a realistic simulation of the physical world. Examples are Distributed Ledger Technologies (DLTs) or Multiagent Systems (MASs). Comprehensive system tests of these technologies require, apart from the simulation of the network, realistic (real or virtualized) network and computing resources and the availability of realistic data or data streams. Virtual commissioning (VICO) using Software-in-the-Loop (SiL) or HiL simulation is already the state of the art for control applications run on Programming Logic Controllers (PLCs) [11]. This paper focuses on the further usage of the simulation of industrial components or plants and presents different use cases and approaches to combine this type of simulation with edge computing. We propose additional usage of existing simulation models for software tests concerning edge computing algorithms and hardware tests concerning highly loaded edge devices. As described in [12], the advantages of VICO for control systems are manifold. They can also be transferred to applications in the context of edge computing in the IIoT, so that the following benefits result:Advance of verification in early stages of implementation;Safety: no risks if systems, i.e., automation technology infrastructure, fail;Security: test setup for cybersecurity testing (e.g., penetration tests);Cost and time savings during development and commissioning;Improvement in quality;Enabling machine learning applications;Test setup for both the logic and middleware layer at the same time;Test setup for both hardware and software.

To the best of our knowledge, there is no existing simulation-based approach for the VICO of data stream systems in the IIoT that takes both networking and computing realism into account, while using plant models of the existing engineering process.

The main contributions of this article can be summarized as follows:Perspective on the evolution and next generation of DSPSs;Description of the usage of simulation from an OT perspective for IT applications in IIoT and the possibility for VICO in distributed systems;Comprehensive overview of possible use cases using physical simulation in IIoT;VICO of a distributed system or industrial application in an experimental setup;Evaluation of feasibility and quality of statements.

The remainder of this article is organized as follows: Section 2 describes the relevant background about the current developments in the fields of IIoT, edge computing, data stream processing, simulation, and standard development processes. Section 3 describes the state of the art of simulation for edge computing and DSPSs in IIoT. In Section 4, the boundary conditions, possible architectures, and application fields are presented. This is followed by a comprehensive, but not exhaustive, collection of use cases in Section 5. Then, an exemplary HiL-coupling is presented and different evaluations are performed in Section 6. The results are discussed and limitations are pointed out. The article concludes with a summary of the main aspects and outlines further research needs in Section 7.

## 2. Background

### 2.1. Industrial Internet of Things Network and Industry 4.0

The term *IIoT* is not uniformly defined. We use the definition from [13] that describes IIoT as a “system comprising networked smart objects, cyber-physical assets, associated generic information technologies and optional cloud or edge computing platforms, which enable real-time, intelligent, and autonomous access, collection, analysis, communications, and exchange of process, product and/or service information, within the industrial environment, so as to optimise overall production value.” The main differences with the Internet of Things (IoT) in general are as follows:The network participants are predominantly industrial devices, e.g., PLCs, iPCs, and higher-functionality field sensors;The type of applications that are run on the devices;The stricter requirements with regard to safety, security, latency, and location awareness [14].

Becerril and Prasanna [15] highlight closed-loop data modeling as a relevant feature of IIoT, as it enables autonomy, e.g., in decision making or work procedures.

According to [14], industrial communication systems can be classified by four criteria:Real-time behavior: non, soft, hard or isochronous real-time;Distribution: higher distributed networks due to wide-area networks (WAN) in addition to existing local-area networks (LAN);Heterogeneity: heterogeneous infrastructure that can also include homogeneous or standardized components;Installation: wired or wireless (e.g., battery-operated devices).

In the context of IIoT, the two standard models, *Automation Pyramid*, specified in the standard DIN 62264 [16], and *Reference Architecture Model Industry 4.0 (RAMI 4.0)*, specified in the standard DIN SPEC 91345 [17], are to be mentioned. While the functional description of the automation pyramid still applies, the RAMI 4.0 is to be seen as an extended model that relates the hierarchy levels (see standard DIN 62264 [16]) to the business process layers and the life cycle stream (see standard DIN 62890 [18]) in I4.0. The existing hierarchy of Industry 3.0 is adapted to the vision of fully connected interacting devices and cross-level communication. The changes from Industry 3.0 to I4.0 are illustrated in Figure 1.

A challenging aspect to highlight are the dynamic changes in the IIoT that become amplified with the described hierarchy changes. Dynamic changes occur, for example, in the network topology when devices or machines are disconnected or added, arranged differently in the production lines, or are subjected to process changes. Enterprises changes in production units and volume, impacts of critical supply chains, and rapidly changing market demands play an even more important role in daily business. It is estimated that in the context of the modularization and flexibilization of production in I4.0, this will be much more frequent. Apart from dynamic changes in network topology, changes in the streaming data itself, e.g., changing number of sources, and in the analysis tasks, e.g., changes in the algorithms or number of tasks, need to be considered.

### 2.2. Edge Computing

The term *edge computing* denotes the data processing close to the users and data sources. It is characterized by its distribution, low latency, location awareness, and limited storage and computing resources [20,21]. Edge computing is of increasing relevance in data processing, as it can overcome the shortcomings of cloud computing mentioned in Section 1. Nevertheless, various challenges stand in the way of continuous deployment. Due to the requirement of economic efficiency, in many cases, the existing infrastructure is to be used. Thus, computing and communication resources are limited. The edge devices are heterogeneous and topology is distributed. In the context of data stream processing, (near) real-time processing is required. In IIoT, the short cycle times are particularly challenging. At the same time, there are various requirements such as the demand for sustainability, scalability, and extensibility. Despite their complexity, edge computing systems should be easy to install and to maintain. In the increasingly digitized and anonymous world, the demand for trustworthiness is increasing [22].

As described in [20], the usage of both edge and cloud computing can cover most of the requirements of the different processing tasks. Depending on the requirements of the respective application, it is to be decided where to execute it the best. The approach of edge and cloud can be seen, for example, in the current development of the virtualization of industrial devices such as PLCs [23,24,25,26].

### 2.3. Evolution of Distributed Stream Processing Systems

As the importance of data processing in industry increases, so does the use and importance of DSPSs. The described trend of edge computing is noticeable in the current developments of DSPSs. While in [27], the evolution of the first four generations of DSPSs is described, this work provides an outlook on the next generation. In the first three generations, the transformation from the origin of data bases and data queries based on SQL to cluster environments and user-defined functions takes place. These systems are deployed in data centers or in the cloud. In 2017, the first attempts were made to integrate edge devices for data preprocessing to solely the cloud, or data center-based solutions so far, which marks the beginning of the fourth generation. In [28], the changes from data bases to data stream management systems and the development in scalable data streaming are also highlighted, but the current trend in edge computing is not considered. We present the vision of a fifth generation of DSPSs that focuses on the data analysis of edge devices and only optional further computing resources in the cloud.

### 2.4. Simulation

Simulation is a widely used term, but it is hard to delimit. According to [29], simulation is the process of developing and running a model of the real system in order to gain knowledge. Leng et al. [30] define simulation as “a technology that simulates the physical world by transforming a model containing deterministic laws and mechanisms into the software. As long as the model is correct and fed with complete context information and environmental data, it can accurately reflect the characteristics and parameters of the physical world.” It should be noted that in this definition the correctness and completeness of a simulation model heavily depends on the use case, e.g., with regard to dynamics, real-time requirements, or the admissibility of a reduction in order. The same applies for the choice of a suitable kind of simulation. A general classification of simulation is made in [31] by the following criteria:Static vs. dynamic simulation (with regard to the occurrence of time-dependent changes);Physical vs. mathematical simulation;Manual vs. computer simulation;Deterministic vs. stochastic simulation;Continuous vs. discrete simulation;Terminated vs. nonterminated simulation;Stationary vs. nonstationary simulation (reaching a stationary phase after a finite time);Real-time vs. non-real-time simulation.

In this work, the definition according to the guideline of the Association of German Engineers VDI 3633 [32] is used. It defines simulation as “representation of a system with its dynamic processes in an experimentable model to reach findings which are transferable to reality; in particular, the processes are developed over time”. The explicitly named time dependency excludes static analysis from this definition of simulation [33].

In [29], dynamic simulation is further differentiated according to the time progress:Time-orientated simulation with fixed time cycle for updating the model;Process-orientated simulation, where simulation time can pass between the call of a method of an object and the return from the call (wait function);Event-orientated simulation based on event sequences (events are not time-consuming).

In industrial contexts, simulation is part of modern development processes. Examples for the most relevant simulations of technical systems are plant simulation, logistic simulation, process simulation, system simulation, logic simulation, multibody simulation, finite element/ volume method simulation, and control loop simulation [34].

As we propose the continuous usage of (existing) OT simulation tools and models for cost-efficient and realistic validation and verification of edge computing and middleware algorithms, as well as VICO, our focus lays on dynamic, computer-aided simulation of the target system, i.e., plants or machines. Depending on the application and the test specifications, real-time requirements may apply.

### 2.5. V-Model for System Development and Virtual Commissioning

According to [35], the V-model describes the development process in three phases: (1). decomposition into smaller units, (2). implementation, and (3). integration of the elements into the overall system. During the integration stage, both verification against the specifications during decomposition stage and validation against the stakeholder needs are to be performed. Initially, the V-model is a model for classic software development, and it was already mentioned in 1984 [36]. In guideline VDI 2206 [37] of the Association of German Engineers, which was updated in 2021, it is applied to the development of mechatronic and cyberphysical systems (CPSs) (see Figure 2) and thus also of relevance for industrial applications.

The guideline applies the V-model for software, electronics/ electrics, mechanics, and other disciplines to all three parallel development processes, i.e., modeling and analysis, and implementation and requirements management. The V-model is a state-of-the-art model for system engineering [38,39]. As the V-model focuses on validation and verification during the engineering process, for control software, it is often linked with VICO, e.g., in [40,41].

In VICO, concepts for validation and verification are based on simulations that enable advanced integration tests before the real plant is build. For VICO of control software, Model-in-the-Loop (MiL), Software-in-the-Loop (SiL), and HiL are described in guideline VDI 3693 [11] and are state-of-the-art. While MiL is the most abstract representation and only requires an automation model, SiL is already based on the correct programming language, i.e., IEC 61131 code [42], and HiL requires both the correct hardware and programming language. The description in guideline VDI 3693 is limited to control units and IEC 61131 code.

## 3. State of the Art

In this section, we first give an overview of the usage of HiL and SiL in the IIoT, then summarize existing approaches for simulation for DSPSs for edge computing. Subsequently, we outline the objectives of this article based on related work.

### 3.1. HiL and SiL for Edge Computing in the IIoT

Liu et al. [43] point out that simulation is a purposeful way to provide test and training data for machine learning models. Simulation in terms of network simulation has already been used many times for validation and testing, e.g., refs. [44,45,46]. In [47], simulation is used for the generation of data sets for machine learning application. In contrast to our approach, the generation of the data and the training process are decoupled in time. Thus, no feedback loop exists. In the context of Reinforcement Learning (RL), the coupling with a simulation model in the sense of agent-in-the-loop is already state of the art, e.g., refs. [48,49,50], but the term *simulation* must be defined more precisely here. Rarely is it a replica of a real system, but instead a highly abstracted simulated training environment such as the open-source python libraries OpenAI Gym [51] or Stable Baselines3 [52]. In [53], a test environment for edge-based CPSs based on HiL is developed. Their model-based simulation is designed for simulation of vehicle dynamics using the SUMO traffic simulator that represents a “microscopic, space-continuous, and time-discrete traffic flow simulation” [54]. HiL is only used for test and validation purposes. The demonstrator is not designed for industrial use cases and the application to DSPSs. Lv et al. [55] present a HiL simulation for the evaluation of the communication of robot control devices. The work extends the traditional usage of HiL for the commissioning of control unit applications by the evaluation of wireless communication. Using simulation in the context of communication evaluation that is not limited to wireless networks or robot systems, may be another relevant research field. The review of Leng et al. [30] confirms the relevance of simulation for I4.0 with a special view on digital twins. Firstly, they highlight the relevance of VICO and HiL in the traditional way for the commissioning of control units. Secondly, they describe the usage of simulation for the building and commissioning of digital twins. However, only examples using discrete event simulation are listed.

In contrast to the previous mentioned related work, in [56], a data-driven simulation approach in manufacturing context is proposed. However, their work focuses on supplier selection and risk management. This is possibly a further field of application for simulation, but it differs from our proposed usage of simulation w.ith regard to the edge computing devices and the loop between simulation and hardware resp. software.

In the context of robot applications, Aldegheri et al. [57] developed a toolchain using HiL for the verification of functional and real-time constraints of robot systems with a 3D simulation. Their robotic software consists of cloud and edge applications, among others AI applications, based on ROS that are orchestrated by Kubedge. While latency is part of the verification, the edge device resources are not considered. The simulation is only used for verification purposes, but not during the software development process.

In [58], the deployment of artificial intelligence (AI) on control units in the field of autonomous vehicles is described. Grigorescu et al. propose a prototyping procedure split up into SiL prototyping executed in the cloud and then transferred to the edge device for HiL testing. In this approach, only one device is in the loop, and there is no focus on industrial use cases or DSPSs.

### 3.2. Simulation for DSPSs of the Fourth and Fifth Generation

This section summarizes the related work about the applications of simulation for distributed systems in the IIoT. The focus of the literature research lies on the DSPSs of the fourth and fifth generation. The results of the systematic literature research with the Google Scholar full-text search confirm the novelty level of the subject (see Table 1). When searching for DSPS, a distinction has to be made between *edge* and *edge computing*, as the term *edge* is used with different meaning. As the stream processing queries that represent the IIoT application are often based on directed acyclic graphs (DAGs), the structure is described with the relation of *edges* and *nodes* [59]. Reducing the search space from IoT to IIoT also leads to significantly fewer results. Additional restriction of the search space confirms the lack of literature about the idea of system development, validation, and verification using typical simulation models of the industrial engineering process.

Liu and Buyya [10] mention in their review article only two simulation approaches, namely the Yahoo Streaming Benchmark [60] that integrates a simulation for advertisements and campaigns as well as a benchmark using video stream data [61]. In summary, Liu and Buyya [10] criticize the lack of realism in performance evaluation as well as the low suitability for IoT.

In [62], the latency of real devices, as well as the results of a simulated video processing, are recorded and subsequently imported in the network simulator NS-3. In this manner, more realism regarding the input data is achieved. However, in this two-step procedure, direct evaluation of the behavior in a real-world setup and in-the-loop prototyping are not possible.

Mudasser et al. [63] build an adaptive fault-tolerant strategy for latency-aware IoT application. Thus, the focus of the evaluation lays on the end-to-end latency in the presented distributed edge computing application. The simulation tool used is iFogSim [64], which is based on DAGs and said to be one of the first edge computing simulators [65]. It is associated with the long-established CloudSim framework that is used analogously to iFogSim for simulating cloud computing systems.

Unlike our approach, existing simulations for DSPSs simulate the network itself, i.e., only the network perspective is considered. The lack of realism due to the pure network simulation, e.g., as DAG in iFogSim, is emphasized in [65]. To solve this issue, in [65], a testing platform based on edge device emulation is proposed. It is suited for container-based and virtual machine emulation. The network emulation can be further extended through interfaces for simulation and physical devices. Unfortunately, the simulation interface is limited to network simulation. Thus, EmuEdge is a more realistic form of network simulation, but it also does not take into account the target system, i.e., plants and the physical environment with their real data streams, and is limited to the network.

In contrast, we propose the usage of simulation models of the later target system, i.e., the entire production facility or machine, not just the network. The pros and cons of the presented related works are summarized in Table 2.

### 3.3. Novelty and Objectives

The novelty of the proposal of this work is the combination of traditional OT simulation methods, i.e., physical simulation of the industrial target system, with new emerging IT technologies such as DSPSs, machine learning, and edge computing. The main features of our approach, which differ from the related works, are as follows:Physical simulation of the industrial plant instead of network simulation;Focus on deployment in IIoT and I4.0;Direct interaction (exchange of data streams) between simulation and application—no two-step procedure;Usage of simulation is not limited to verification and validation but considered during the whole software development process;This article provides a holistic overview of the wide range of use cases for this approach. It is not limited to a single implementation;Application of HiL and SiL is not limited to a single device; it is for entire systems;Higher degree of realism.

According to [65], the following aspects are to be considered when creating “realistic and reproducible edge computing experiments”:
•   Link realism•   Topology flexibility•   Traffic realism•   Scalability•   Resource realism•   Easy replication•   OS realism•   Low cost•   Functional realism

For the IIoT context and industrial use cases, we place the loop between simulation and software or hardware as an additional aspect.

In [65], the term *Degree of Realism* is defined to describe the “degree to which an emulation system mimics real-world performance”. In [65], this is limited to the computation and networking performance. In our work, we add a third criteria, the realism of data and data streams generated by the simulation model.

The proposed usage of simulation is intended to extend the state-of-the-art approaches for validation and verification procedure that use network simulators only.

**Table 2 sensors-23-03545-t002:** Summary of pros and cons of related work.

Related Work	Application Field	Pros	Cons
HiL and SiL for Edge Computing in the IIoT			
[47]	machine learning	+ use of simulation for data generation	- no streaming data, collection of data sets - no direct interaction, no feedback loop - simulation and training are decoupled in time
[53]	vehicle dynamics	+ test environment for edge-based CPSs + HiL + model-based simulation	- only used for test and validation purposes - not used during software development process - no focus on industrial use cases or DSPSs
[55]	communication, robotics	+ evaluation of robot controller communication + HiL	- limited to communication testing - hardware resources are not considered - focus on robot platforms operated in cloud - no focus on edge computing, DSPSs or AI
[30]	digital twins	+ confirms the relevance of simulation for I4.0 + confirms the relevance of classic VICO and HiL + simulation for development of digital twins	- limited to discrete event simulation
[56]	supplier selection, risk management	+ simulation combined with machine learning	- no focus on edge computing or DSPSs - limited to discrete event simulation
[57]	robot system	+ verification of functional constraints and latency + physical simulation + machine learning application + deployment in cloud and edge	- no feedback loop - no consideration of edge device resources - simulation is only used for verification purpose - simulation is not used for software development
[58]	autonomousvehicles	+ machine learning on control units + SiL and HiL + prototyping in cloud and edge	- only one device is in the loop - no focus on industrial use cases or DSPSs
Simulation for DSPSs of the fourth and fifth generation			
[10]	DSPSs	+ criticizes the lack of realism in performance evaluation in state-of-the-art approaches + criticizes the low suitability for IoT	- no solution provided (review article )
[62]	stream processing	+ real-world values are recorded + realism regarding the input data	- discrete event simulator - two-step procedure; thus, no feedback loop - decoupled in time - evaluation in a real-world setup not possible
[63]	IoT	+ evaluation of end-to-end latency + distributed edge computing application + edge computing simulator	- based on network simulator
[65]	Edge computing	+ identifies a lack of realism in the state of the art + container-based and virtual machine emulation + interfaces for simulation and physical devices	- only network simulation interface - no consideration of target system - is limited to the network

## 4. Method

This section describes the features as well as possible architectures of the proposed method. It narrows down the simulation types in order to fulfill the requirements of the industrial context. The content of the approach is mapped to the general structure of DSPSs.

The focused application area is I4.0, especially factory automation, but it is not limited to this. Due to the industrial context, some boundary conditions need to be considered. The infrastructure layer consists of a high number of connected industrial devices in a production area, such as an industrial plant. Cycle times are high and soft, or hard real-time requirements may apply. The installation of devices is traditionally wired, but wireless installation, e.g., wireless sensor networks, increasingly occurs. The heterogeneity of devices is to be considered, but communication standards such as network protocols or the asset administration shell [66] exist. Many devices are located at a fixed location, but also mobile devices exist that move within a predefined area, e.g., automated guided vehicles. Common fields of industrial application are condition monitoring and predictive maintenance of the plant itself, as well as quality improvement and early product defect diagnosis.

There are different points in time (see Figure 3) during the life cycle of a plant, when the described use of simulation brings benefits. Additionally, if another new plant is installed in the same factory, the interconnection between the devices is another relevant task.

The main features of the proposed approach are as follows:Transfer of the standard procedure of VICO for control loop applications to edge computing applications.VICO for systems in the IIoT that are not limited to a single algorithm on a single edge device but complex interaction in a system running in an industrial environment.Usage of traditional OT methods for realistic and cost-efficient VICO, as well as validation and verification of IT applications in the IIoT.

As this work proposes the usage of traditional OT simulation models and tools, the simulation is to delimit from, e.g., IT simulations such as network simulators based on discrete event simulation. The considered simulation types and simulation tools, respectively, in this work meet the following requirements:Generation of realistic data streams. The simulated streams should resemble the real data and data streams at the manufacturing sites;Dynamic, time-dependent simulation, often time-orientated simulation, i.e., with hard cycle times, is required for realistic validation;Physical simulation of the whole target system, e.g., components, plants, etc., in contrast to network simulation limited to nodes and edges;Applicability for HiL and SiL;Real-time capability for time-critical applications.

In many cases, the simulation model already exists, as it is developed during one of the previous development and design process stages in the fields of mechanical engineering or electronics, which leads to high realism and cost-efficiency.

As the application of simulation is especially proposed for stream processing systems, the content of this work is transferred to the architecture of DSPSs. According to [10], data stream systems generally have a three-tier structure: the logic layer, the middleware layer, and the network-layer. While the logic layer contains the edge computing algorithms for data processing, the middleware layer is built from the DSPS itself, as well as enabling software for DSPSs. The network is represented by the simulation. In the case of SiL, the real system can be fully replaced by the simulation model. In the case of HiL, the target system is also represented by the simulation, but some or all edge devices exist physically or are virtualized and coupled to the simulation. The content of this work mapped to the three layers of data stream systems is shown in Figure 4.

For the proposed SiL or HiL setup, different architectures are conceivable, but all have two features in common. Firstly, a coupling to a simulation model is established. Secondly, data are generated during simulation and are transmitted to the coupled device(s) or software for processing. By *data*, on the one hand, is meant continuous data streams such as simulated sensor signals and, on the other hand, single data such as events.

In this work, we define SiL and HiL as follows.

**SiL** is limited to software tests, as it has no relation to hardware. SiL requires the applications to evaluate, e.g., the edge computing algorithms of the logic and middleware layer of the data stream system, to be in the same programming language as in the (real) target system but independent from the later executing device. This piece of code can be run on the same computer or, depending on the capabilities of the simulation tool, directly in the simulation.**HiL** is suited for both software tests and hardware-dependent tests. The algorithms are running on the target device, in the correct programming language, and installed and deployed as in the future target system. Thus, the discrepancies to operation in the real (target) system are minimized. We also explicitly assign virtualized hardware in the sense of virtual machines or docker container to HiL.

In contrast to [11], in this work, we propose the simulation coupling for general edge computing tasks and not only control loops. Thus, the possible programming languages are not limited to the IEC 61131-3 standard [42].

There are several simulation environments from different fields with different targets that could be suitable for the described use cases and requirements, e.g., MATLAB/Simulink [67], iPhysics [68] or simulation tools with, e.g., an FMU standard [69] conform model export such as Dymola [70], so that the model can be processed by suited HiL simulators. Similar criteria need to be taken into account for the VICO of control units. In general, the coupling depends on the requirements of the specific application. One can distinguish between the following:Real-time or non-real-time simulation;Testing a single device or several networked devices;Testing a single algorithm deployed on one device or on a whole system;Communication in the loop or unidirectional;Communication to device (HiL) or software (SiL);Realism of the transmission medium.

In this work, we present three possible architectures. In addition to the common HiL setup with only one hardware device connected to one simulation model, the system approach is focused on.

### 4.1. Architecture 1

Figure 5 illustrates one simulation that is coupled with one edge device that is optionally connected with other networked devices, e.g., real, virtualized or simulated devices. At this point, the possible connection to frameworks such as the real-world network simulator EmuEdge [65] is to be highlighted. EmuEdge is mainly based on emulation but has interfaces for network simulators and physical devices, which increases scalability.

### 4.2. Architecture 2

The second proposed architecture is a single simulation that is connected to multiple edge devices, as shown schematically in Figure 6. For each device, a separate communication is established.

### 4.3. Architecture 3

The most complex architecture consists of several simulations, each connected to one or multiple edge devices. The simulation models can either be homogeneous, i.e., the same simulation model is multiplied and synchronized, or heterogeneous, e.g., representing different plants in one factory, as well as the communication between the edge devices of different plants. Figure 7 presents an exemplary representation of the third architecture, whereby the network structure (mesh net), number of linkages and nodes, etc., are chosen exemplarily and can appear in different variations.

In the described architectures, the *Loop* between edge devices and simulation is not obligatory—in contrast to state-of-the-art HiL for control loop applications. While the transmission of data and data streams from the simulation model to the edge device resp. software is required, the transmission of the feedback and processing results is optional, but it is relevant for some use cases (see Section 5.2.3 and Section 5.2.9).

In this study, an exemplary HiL implementation for a DSPS and DLT network according to the second architecture is realized (see Section 6).

## 5. Comprehensive Overview over Possible Use Cases

This section gives an overview over possible use cases, where the application of simulation in the IIoT environment generates additional benefits. The focus lays on the usage of physical simulations, possibly combined with visual simulation (3D models), which are already a state-of-the-art part of the engineering process for all factory levels. The first Section 5.1 shows, by means of three exemplary systems, the relevance of simulation in system development and the VICO of systems. The second Section 5.2 presents general use cases in the context of edge computing in IIoT that are not limited to the system character.

### 5.1. Simulation for System Applications

A system consists of multiple network participants that are connected and interact with each other. Three exemplary kinds of emerging edge computing systems in IIoT, i.e., DLT, MASs and DSPSs, are presented and the use of simulation described. For all systems applies that simulation helps to verify and test on the following scenarios:
SiL:System test for validation of the software functionality;Dynamic adoption of the system when adding further network nodes;System behavior in case of removal of network nodes or agents.HiL (additional to SiL):Sizing of the hardware;Behavior in case of overload;Behavior in case of hardware errors or defects.


#### 5.1.1. Distributed Ledger Technology and Smart Contracts

Although the benefits of DLT for IIoT are highlighted several times in the literature [22,71], DLTs are still not common in today’s factories, as they are often associated with high computational load. Nevertheless, the convergence of DLT and IIoT is to be expected, and research is being conducted on technologies focusing on the constraint resources of IoT and IIoT devices. Especially, DLT IOTA is to be mentioned in this context [22,71,72], among others, because of its low computational overhead. IOTA is even in discussion to become a machine-to-machine communication standard, according to [73].

In addition to the advantages of DLTs themselves, they enable another relevant technology: so-called Smart Contracts (SCs). To validate the behavior of SCs, the availability of realistic data is needed. As SCs are triggered by the fulfillment of a condition on the distributed ledger, they might depend on the realistic input of the simulated plant or have influence on it. For validation of the interaction and dependencies, realistic data resp. data streams are needed, e.g., provided by simulation.

In addition to the general verification of the DLT network, e.g., in terms of difficulty level of consensus algorithms, computing resources for different node types, etc., further usage is to be demonstrated by means of two examples. Considering the use cases of DLT in IIoT described in [22], the verification of DLT and SCs depends on realistic plant data:Immutable, long-term storage of selected relevant IIoT dataFor various scenarios, e.g., legal requirement of proof, the long-term storage and verifiable immutability of selected data are necessary. Different to traditional manual paper-based records, DLT enables a digital and automated record. Data streams and event data can be generated by the simulation, and the verification of the functionality—in the case of IOTA the distributed data base SkyllaDB and the permanode functionality—can be run. This can clarify how much additional load is put on the devices, which devices are suited for running the distributed data base and DLT nodes, which scope of data (data rate and amount of data) is realistic to be stored, and so on.SC-based access right managementSCs are digital contracts characterized by more determinism and automation. Digital services, pay-per-use business models and access right management, e.g., for reading and writing IIoT data and analysis results, are only some of the possible use case of SCs. In the aforementioned case, a customer pays for a digital service and receives access to the respective data or information for a certain period of time, e.g., results of an anomaly detection, condition monitoring or predictive maintenance. With simulative analysis, the correct execution of the SC and the verification of the created SC can be tested, respectively. In addition, the latency between fulfilled condition and execution of SCs can be measured and experiments with regard to the optimization of quorum and choice of nodes, e.g., depending on their reputation, that decide about the SC.

Using HiL with a DLT network and further DLT-based functionalities allows the evaluation of hardware suitability, e.g., for IOTA GoShimmer binaries for ARM processors that are not yet available, as well as hardware resource usage. With SiL, the system tests and integration tests are possible, and the interaction between the distributed ledger, SCs, etc., and the plant can be evaluated at an early stage. This leads to benefits with regard to advancing DLTs and SC in IIoT, since valid statements about suitability and feasibility can be made independently of a physical plant. In addition, the quality of the SC functionality, DLT system, and a possible overall system is improved due to the possibility for comprehensive tests during the whole development phase.

#### 5.1.2. Multiagent Systems

Agent systems are one of the key technologies in I4.0 [74]. Since the agents interact with each other on the one hand and with their environment on the other, and thus depend on the current state of the environment, a simulation of the future target system is a relevant feature of the system test.

For the validation of agent systems in general, simulation can be used for evaluation of the following:The performance in environment;The interaction between the agents;The behavior of agents in error states;The behavior when devices or agents are added or removed in the system.

Highly relevant fields for agent systems in industry are network technologies, e.g., agent-based scheduling and load balancing [75,76], job shop scheduling [77,78], and logistics [79,80,81]. Depending on the application, the choice of an environment simulation is preferable since, e.g., machine load, production time, transport routes, etc., are also simulated.

A special case of agent systems is RL, a discipline of machine learning, in which one or more agents interact with an environment (agent-in-the-loop) and learn the desired behavior via trial and error. As with all machine learning applications, RL is data-driven, and the agents need to be trained. The training procedure is often performed in a simulated training environment. This simulation generally differs from the described physical simulation and is only used for RL agent training. The other way round, however, a physical simulation can be used as training environment. Applications for RL agent systems in the context of DSPSs exist for both the logic layer, e.g., ref. [82], or middleware layer, e.g., ref. [83].

A SiL setup of an agent system to train with the simulation environment can reduce time, since no additional training environment has to be developed, and the behavior of the simulation model is often closer to the target system than an abstract RL training environment. Thus, the model quality increases and the behavior of the agent in training and execution will be more similar.

#### 5.1.3. DSPS of the Fourth and Fifth Generation

The current developments of the fourth generation of DSPS and the expected fifth generation show the relevance of edge devices for data processing (see Section 2.3). As a further stage of verification, subsequent to or in combination with network simulators, the physical simulation can provide a more realistic simulation of the target system. In accordance with Figure 4, the simulation has an influence on the representation of the infrastructure layer. The plant is partially or completely replaced by the physical simulation. For the middleware layer and logic layer, no difference to a real setup should be recognizable. While for the verification of the middleware layer, the setup and infrastructure are relevant, the verification of the logic layer highly depends on the data themselves.

An existing trend is to develop software independently of the hardware deployment and only afterwards choose the best location for processing. In the context of virtualized edge devices, the dynamic allocation of resources for control tasks or other stream processing tasks, especially under consideration of latency and real-time requirements, is one of the key questions.

In addition to the decision about the deployment on virtual or physical devices, another relevant use case is the optimization of task distribution on networked computing units with given requirements, such as hard real-time, permissions, priority, and available resources. The usage of simulation in this context is explained by means of an agent-based resource allocation for logic-layer algorithms in DSPSs, as proposed in [83]. Agents decide about the available resources or routing to neighbors. The start and execution of the logic-layer applications and the resulting resource usage, concerning both computing resources of the devices and communication resources of the network, as well as latency and results, can be directly tested in a HiL/SiL setup. Thus, it provides more extensive and complete tests, since both the middleware and logic layer can be evaluated together. This leads to improved tests of functionality and robustness, as well as system and integration tests. Additionally, the realistic setup simplifies the transfer to the real plant.

Therefore, the use of physical simulation for DSPSs design and development has the advantages of improved quality, as well as cost and time savings due to VICO in advance of physical commissioning.

### 5.2. Simulation for Edge Computing Algorithms in General

Regardless of the deployment in a system or on a single device, usage of physical simulation is beneficial for edge computing algorithms in general. The following presented use cases give a comprehensive, but not exhaustive, overview of possible application scenarios in the industrial context.

#### 5.2.1. Prototyping and Choice of Algorithms and Software

In the design stage, different algorithms are often discussed regarding their suitability for an intended application. Thus, testing and comparing prototypes under realistic conditions in the early stages of the development process and far before deployment in the real plant is of interest for the best decision making. Simulation can be used for the comparative analysis of different algorithms with the same simulation models of a machine or plant under the same conditions. With regard to the different stages of the software development life cycle, simulation can find usage in almost every stage, from design to the maintenance of software. For this use case, SiL is the preferred method. The goal is to find the most suitable algorithm resp. machine learning model and parameter values, which has the benefit of improving quality.

#### 5.2.2. Choice and Sizing of Hardware for Edge Computing

The edge devices have limited resources in regard to computational power, memory and, in the case of battery-powered devices, energy. Due to cost-efficiency, hardware is in general fitted to the planned tasks and is not oversized. Thus, experimental evaluation of the capabilities of classical OT edge devices, e.g., control units, for additional IT tasks are necessary for the choosing and sizing of hardware. Examples for metrics of interest are resource load, achievable cycle times, and latency.

If existing hardware is to be used for additional edge computing tasks, tests must be carried out beforehand. Instead of trial-and-error testing in the field, a device of the respective hardware can run its main application, e.g., PLC tasks for control units, in the loop with a simulation, while additional computing tasks are added. For this use case, HiL is proposed to test different hardware and/or algorithms with different parameterizations. In this way, a realistic statement can be made as to whether the hardware resources are sufficient for the planned tasks. For some algorithms, a trade-off between accuracy of results and computational intensity exists, and simulation helps in the search for the optimum. This benefits in saving costs, as hardware is not oversized, in improved quality, and in enabling edge computing, as it is an option to test relevant hardware such as AI hardware accelerators or different processors in a realistic way. A lean hardware setup also shows benefits in regard to power consumption (important not only for battery fed systems) and the sustainability of the new generation of widespread IIoT technology.

#### 5.2.3. Generation of Optimal Data for Training Machine Learning Algorithms

For machine learning, i.e., data-driven algorithms, the availability of data for training, validation, and testing is crucial. There are several scenarios where real world data is not available and for which simulated data could be a suitable alternative.

There are two major problems that can be solved by the usage of physical simulation:Qualitative aspects: The quality of machine learning models is highly linked to data quality. From a quality point of view, simulation enables machine learning in industry as follows:–Not for each possible scenario that can happen in real production processes—data are recorded and available. In simulation, the full spectrum of scenarios, including destructive or dangerous events, can be manually generated and data recorded or streamed;–Exclusively good data and correct labeled data: strange effects due to unknown influencing factors do not appear accidentally in simulation; Training with abnormal data or wrongly labeled data leads to models of worse quality;–Data generated in simulation allows reproducibility.Quantitative aspects: For data-based learning, a certain amount of data needs to be available for training, validation, and testing during model development and evaluation. There are different reasons that lead to a lack of available data in industry. As edge computing is, among others, motivated by data security aspects that are at the same time one of the biggest disadvantages in cloud computing, processed data streams in industry are often only available locally in the plant, and thus not accessible to data science engineers during development. Another scenario occurs in the early stages of the engineering process of a new component or plant itself, i.e., the component or plant does not (yet) exist. Furthermore, the value of data was not recognized for a long time, and historic data were not saved at all or without the required metadata. Moreover, in brown-field applications (retrofit), sometimes the plant exists already, but the required sensors are installed along with the commissioning of the processing algorithms.

A special case of machine learning training is RL, which is characterized by an agent that acts in a loop with its environment. Analogous to the described multi-agent use case in Section 5.1.2, the usage of simulation is not limited to data generation but is even used for RL agent-in-the-loop training.

The proposed data stream generation can take place with a HiL or SiL setup, or by caching the generated data to a file. The method enables and improves the training, validation, and testing of edge computing algorithms. This may result in the realization of new data-driven approaches, and thus technological advances and improved quality, e.g., through online anomaly detection that detects defects at an early stage. This in turn can lead to cost savings and new business models.

#### 5.2.4. Planning of Sensors and Development of Virtual Sensors

Sensor data have become increasingly relevant today due to the high number of data-driven approaches. Thus, the planning of sensors in a plant is of high relevance. There are two ways to capture sensor signals. The traditional way is the direct measurement via physical sensors. A new research field in the context of data streams is virtual sensors. In contrast to physical sensors, the signals of virtual sensors are indirectly captured, i.e., virtual sensors are used to process and fuse one or more signals from physical sensors to generate a new signal [84].

In a first use case, simulation can be applied when planning sensors in a machine or plant during the development stage, including the identification of necessary physical sensors and possible virtual sensors. A second use case is the development and testing of virtual sensors by means of simulation. Different methods for virtual sensors, among others neural networks, are used [85,86]. Thus, if virtual sensors are identified, training of the algorithms is required. This can be carried out in advance during development stage, before the real plant is built, using sensor signals from simulation in a SiL setup.

This allows to save costs as only the required sensors are installed as well as retrospective installation of missing sensors can be avoided. It enables data-driven approaches by allowing sensor requirements to be considered at an early stage. In addition, the quality of virtual sensors can be improved during the design stage and training of the virtual sensors on the simulation model.

#### 5.2.5. Hyperparameter Optimization

Hyperparameter optimization (HPO) can significantly improve the performance of algorithms. Instead of a manual search that was the state of the art for a long time, automated search algorithms are used. This is based on the search for optimal hyperparameter values and their combination, while reducing the value of a loss function. Depending on the use case, applying HPO in the loop with a simulation model offers new possibilities [87]. As simulation can provide a high degree of realism, the hyperparameters of various kinds of algorithms that are directly linked to the plant can be optimized. Examples of these algorithms are process optimization, material flow, and the routing and interaction of automatic guided vehicles. In general, SiL is proposed for HPO, but HiL is required if the hyperparameters to be optimized also depend on hardware resource usage. The same applies for decisions about real-time or non-real-time simulation. If time is relevant for optimization, e.g., in terms of latency, real-time simulation is needed. HPO is directly linked to the improvement of quality and efficiency.

#### 5.2.6. Providing a Ground Truth

In data science, it is not always obvious which values are the correct ones. Thus, a ground truth is asked for. The results of simulation are a possible answer. This is not limited to the development process of algorithms but can be applied in the operating system as well. The simulation runs parallel to the monitored process in the plant, which is closely related to the research field of digital twins. Possible applications for the parallel simulation are anomaly detection or data compression.

Anomaly detection: Anomalies can be detected by comparison between the behavior of a simulation model and the real world. Differences that are above average indicate anomalies. Different kinds of anomalies can be detected this way, e.g., drifts, i.e., slowly emerging anomalies, which are characterized by an increasing difference over time or point anomalies that are single outlier values.Data compression: In delta compression, only the difference between predicted value and real data is transmitted. For lossy compression, the efficiency can even be increased by reducing the transmission to differences above a certain threshold.

The examples show that a ground truth is the basis for some data processing methods. Parallel simulation can provide this ground truth and enables these methods for I4.0, and thus for additional digital services. This research field is closely related to digital twins.

#### 5.2.7. Testing Faulty States

During the lifetime of a plant, faulty operating states can occur and might even be accompanied by damage to the plant. This is the reason why alarm and error handling is a core functionality that must be covered by any VICO approach. Within the framework of a simulation, various problems can be generated, and the robustness of the algorithms can be tested against them. Thus, comprehensive tests that exceed the possibilities on a real plant are possible due to the option to generate faulty states directly in the simulation. On the other hand, testing prototype software in a simulated environment is of low risk, as failures in software or overloading the edge devices do not represent a danger for the plant itself, the environment, or humans. Some simulation tools, e.g., iPhysics [68], offer visualizations of the simulated environment, which is a helpful method for developers to understand the reasons for faulty behavior.

Comprehensive testing improves the software quality regarding robustness and functionality. By using simulation, physical resources can be saved, and the scope of the tests is much larger compared with real-world tests, while the risks are minimized.

#### 5.2.8. Dealing with Dynamic Changes

The IIoT is a highly complex network, with the occurrence of dynamic changes as one of its main characteristics. Three kinds of changes are presented in [83]: changes in the network topology, changes in the streaming data, and changes in the data analysis tasks (see Section 2.1). Thus, adaptivity to dynamic changes is a relevant requirement for algorithms executed in the IIoT. From the perspective of edge computing, the suitability of edge computing algorithms is to proof for the following aspects:Changes in topology:–Are the algorithms robust to deal with a dynamic number of network participants (addition or removal of faulty devices)?–Are the algorithms robust to deal with changing linkages between the devices?–Can data loss be avoided and processing tasks be transferred to other network participants?Changes in data streams:–Can the processing algorithms handle changes in sampling rates or additional or removed streams?–Can data loss be avoided?Changes in algorithms:–Can software be updated during operation mode?–Is a continuous execution of the algorithms granted?–Is consistency of software revisions guaranteed throughout the DSPS?

By using simulation for testing the robustness and adaptivity of algorithms and systems to dynamic changes, the software quality can be improved at an early stage and less errors occur during (real) commissioning.

#### 5.2.9. Prototyping and Validation of In-the-Loop Applications

Analogous to control applications, edge computing applications such as data analysis can lead to a result that triggers further steps in the production process or plant. Examples are visual signals, e.g., for pick-by-light applications or signaling machine operators, the detection of an anomaly in the area where it is probably caused, or the optimization of material flow and process routes. Thus, a feedback loop is required for full verification. During the design and development phase, HiL and SiL can be used to verify the behavior in the simulation environment and test the functionality of the application.

#### 5.2.10. Training of Operators and Maintenance Staff

Digitization, e.g., in terms of distributed systems for edge computing, leads to new challenges and more complex activities for employees. Operators and maintenance staff are no longer exclusively responsible for the hardware and classic OT tasks (e.g., control applications) but also for the software running on the hardware. The employee plays a crucial role here, e.g., in the following:Performing updates to the distributed system;Adding or replacing models or algorithms for edge computing in the distributed system (possibly even during run time);Reacting in case of software errors;(Re)starting the distributed system;Starting a new training phase after parameter changes;Evaluation of the results of the data processing, e.g., detected anomalies (is it an indication of a defect and must it be investigated, or are explainable external circumstances responsible for the deviating behavior?);labeling data, i.e., data streams are stored and the operator adds meta-information during the operation, e.g., about currently occurring changes, defects, replacements, etc.

Increasingly complex tasks, such as the operation of fifth-generation DSPSs, lead to the need for appropriate training.

The proposed test bed with edge computing on physical and virtualized hardware, combined with an application-orientated simulation, can provide a suitable training environment for education in these complex tasks.

#### 5.2.11. Marketing and Sales

Since digital services are more abstract than physical components, marketing and sales also have to adapt to digitalization in industry. Simulation is a powerful tool for demonstrations without any physical plants involved. It is suited for better explanation and presentation of digital services combined with a machine or plant model. For example, the added value of digital services can be better demonstrated to customers if it is demonstrated with a visualization of a simulated plant—in the best case already on a simulation model of the customer’s plant. In addition, error states can also be simulated in the simulation, and thus, the advantage of algorithms, e.g., condition monitoring, can be made clear.

Using simulation for show cases of new digital services in the IIoT can create enthusiasm and understanding among customers for the product, e.g., at fairs. Since the success of a new product is directly dependent on the success of the marketing and sales, this use case is of high relevance from a business perspective.

## 6. Simulation Experiments

This section contains the results of the experiments to evaluate the presented approach of using physical simulation for the VICO of data stream systems in IIoT. For evaluation, the exemplary use case of an RL-based MAS for intelligent resource allocation in IIoT with edge computing is examined. It comprises both the MAS as enabler for DSPSs [83] on the middleware layer as well as different data stream processing algorithms [4,22,88] that belong to the logic layer. While the single approaches have already been evaluated individually, the proposed application of simulation is used for a realistic evaluation of the overall system and the interaction of the single approaches in combination with an industrial plant.

The goal of the evaluation is to confirm the mentioned statements about the usage of simulation for edge computing in I4.0. From this, the setup presented below is derived, including physical and virtualized hardware in the loop with a simulation model of an exemplary plant and different interacting algorithms to test.

### 6.1. Experimental Setup

The subsection about the experimental setup is divided into the descriptions of the infrastructure, middleware, and logic layer of the exemplary data stream system in IIoT. It is followed by the experiments and the results.

#### 6.1.1. Description of the Network Layer

The network layer consists of two parts: (a) the networked (physical and/or virtualized) edge devices and (b) the simulated machine or plant (target environment). For the experiments, a network of industrial control units is set up to represent an IIoT network. It is built from physical ctrlX COREs from Bosch Rexroth with a 64-bit quad core ARM A53 CPU, 2 GB RAM, and 4 GB eMMC memory and virtualized ctrlX COREs virtual from Bosch Rexroth with a 64-bit quad core AMD-V CPU, 4 GB RAM, and 4 GB eMMC memory, run as virtual machines on two performance books with Intel Core i7 processors and 32 GB RAM. The operating system is Linux Ubuntu Core 20, which requires all applications to be run as snaps [89]. The network consists of three physical and nine virtualized control units that are connected via Ethernet and a router and a switch (see Figure 8).

A total of twelve control units were selected for the designed proof of concept to ensure sufficient network size for the evaluation of the distributed MAS. The edge devices coupled to the simulation model via HiL serve as access points for the data streams, i.e., as data sources for the edge computing algorithms. For this reason, four selected motion control applications are distributed across four control units to create more than a single source of data streams. As network structure, a grid topology was chosen that is a special type of mesh network.

The second part of the network layer is the connected simulation of an exemplary plant (see Figure 9). Since the experiments pursue the goal to prove the feasibility of the described system, the simulation tool *iPhysics* from the company machineering [68] was chosen. iPhysics is a process simulation with a 3D visualization and fulfills the requirements of HiL coupling. It is based on CAD import and subsequent assignment of properties and parameterization of the individual components. In this first proof of concept, a non-real-time setup is chosen. Nevertheless, the subsequent transfer to real-time simulation is possible with iPhysics, if simulation is run on a real-time target and the iPhysics *FieldBox* [90] is used for field bus emulation.

The selected simulation model represents a palletizing portal packaging plant. Four control units are connected directly to the simulation and control the motion of different elements of the plant. Twelve data streams, further described in Table 3, are generated and transmitted to the IIoT devices.

The PLC projects and the simulation model, as well as the configuration files for the communication between simulation model and control units, can be found in the GitHub repository https://github.com/JuliRosenberger/iPhysicsEvaluation (accessed on 27 February 2023).

#### 6.1.2. Description of Middleware-Layer Application

In this evaluation, the goal is to validate a data stream system consisting of a simulated network layer based on physical simulation using the simulation tool *iPhysics*. In the middleware layer, an enabler technology for data stream processing is examined, i.e., a MAS based on RL intelligently allocates edge device resources for additional data analysis in near real time. Thus, on each IIoT edge device, an agent snap is run that is responsible for resource allocation and decision about execution of additional processing tasks of the logic layer (described in detail in the next section). Furthermore, a snap for monitoring the resource usage, as well as a watchdog snap that intervenes in case of overload, are installed. A detailed description can be found in [83].

#### 6.1.3. Description of the Logic-Layer Applications

Three kinds of high-level edge computing algorithms are run on the edge devices:Anomaly detection:The anomaly detection is based on unsupervised learning, i.e., the nonparametric statistic method kernel density estimation. The efficient one-pass algorithm is suited for the detection of different kinds of anomalies in streaming data (for details, see [4]).DLT and SCs:In a previous work ([22]), we presented two use cases for DLT in the IIoT. The first use case describes the immutable long-term storage of relevant data using the DLT IOTA. The second use case describes a possible access right management based on IOTA in combination with SCs. For both use cases, an IOTA network has to be established on the IIoT devices.Data stream compression:One approach to overcome the bandwidth limitations is data compression. In [88,91], different machine learning approaches for compression of streaming data were considered and compared with traditional methods. In this work, an autoencoder and the traditional algorithm fpzip [92] are executed.

### 6.2. Conducting Experiments and Results

For evaluation, different experiments show the virtual commissioning of the described overall system, including integration tests. An overview of the experiments is given in Table 4.

In the first step, the described setup is used to confirm the executability as an overall system. While setting up the overall system in a realistic manner, several questions arise:The choice between two different implementations for streaming the data from simulation to edge devices that are available for selection. The choice has to be made between a client subscription, i.e., the logic-layer applications subscribe to the respective data streams, or function blocks in the PLC application that forward the data streams to the respective addresses for processing. In this work, it was decided to implement the client subscription. For the sake of completeness, an evaluation and comparison of the second approach would be useful. In the described setup, this is possible to conduct under very similar boundary conditions, thanks to the simulation setup.The second question that was not considered yet is the instantiability of the snaps for edge computing. As several incoming data streams can occur on one device for the same processing tasks, in this use case, several instances of snaps (one snap per stream to be processed) have to run in parallel.Another fact that was not clear before is the demand for normalization of the representation of the CPU utilization of the control units. As one of four kernels is reserved for the PLC applications, the utilization for edge computing can only achieve a maximum of 75%, which is below the threshold of 80% specified in the training.

This first experiment resulted in a successful proof of concept. It shows that the control units can control the simulation plant via HiL, while in parallel, agent snaps allocate free computing resources and trigger data compression and anomaly detection on the data stream of the simulation. In addition, an IOTA 2.0 network of variable size is established. During evaluation of the executability as an overall system using simulation, questions arose that were not considered during the evaluation of the single components of the system without simulation.

Secondly, the agent system of the middleware layer is evaluated with the experiments described in Table 4. The agents performance, i.e., fulfillment of their task to allocate edge device resources, is measured. The measurements confirm the results of the very low resource usage of the agents, as well as the fast decision making presented in [83]. Figure 10 schematically illustrates the overall system.

The agent system is tested on its adaptivity to dynamic changes in the network topology and behavior in case of overload:No errors occurred when changing the number of network participants;When adding devices in the network, the new devices started to process tasks within the scope of their capabilities;When overload of one device occurred, the currently executed computing tasks were returned for new assignment to the further existing devices. However, there is an interruption in the execution of the task until it is assigned to another device. This shows that the agent control leads to a more *stable* execution of tasks, as they are repeatedly assigned in the case of the failure of a device;Starting or stopping the simulation, and thus changes in the data streams for simulating, e.g., network defects, did not interrupt the correct execution of the MAS.

Thirdly, the logic-layer applications are evaluated.

**Anomaly detection** An anomaly was manually initiated via forcing an abnormal value in the simulation environment, which led to a wrong behavior of the machine, as shown in Figure 11. The gripper arm was forced to remain in a low position so that the boxes collided with the gripper arm. The anomaly was correctly detected as a point anomaly by the anomaly detection snap. It is proven that the proposed simulation coupling is suited for evaluating anomaly detection on edge devices.

**DLT** A private IOTA network was built. It consisted of nodes that were run on the networked physical and virtualized ctrlX COREs. It was possible to add or remove participants to the IOTA network during run time without causing errors in the established IOTA network. The distributed ledger, the so-called *Tangle*, synchronized, and transactions were performed successfully.

**Data compression** Two data compression algorithms were executed on streaming data, buffering in sequences of 20 data values for compression. The traditional algorithm fpzip is a lossless compression method. It could be demonstrated that the algorithm compressed and decompressed the data streams from simulation without any reconstruction error. The second algorithm, an LSTM autoencoder, also compressed and decompressed streaming data. The reconstruction error was relatively high, as it was trained on another setup. Therefore, a retraining of the autoencoder coupled with the simulation model is planned in a next step.

### 6.3. Limitations

The experimental evaluation presented is subject to the following limitations:A comprehensive overview over various applications of simulation is given in Section 5. The evaluation is limited to the use cases presented in the Section 5.1.1, Section 5.1.2, Section 5.2.1, Section 5.2.2, Section 5.2.3, Section 5.2.7 and Section 5.2.8;Since the edge computing algorithms themselves have been independently evaluated in our previous works, measurements and evaluation metrics of the experiments conducted are limited to evaluate system character and adaptability to dynamic changes;To evaluate the effectiveness and quality of training machine learning algorithms with a simulation, a comparison with inference with real machine data is required;The simulation is based on iPhysics, i.e., a tool that requires a CAD data import. Other types of simulations and tools may also be appropriate but have not yet been evaluated;Our experimental setup is so far limited to a system according to architecture 2; In further experiments, both architecture 1 (Figure 5) and 3 (Figure 7) should also be tested;The experiments were not yet performed in hard real time, since the simulation was run on a Windows notebook and Python was used as the main programming language;The experiments so far are based on the coupling between ctrlX CORE control units and the simulation environments iPhysics. Additional experiments with other edge devices, simulation tools, and programming languages are necessary for better assessability of the approaches.

## 7. Discussion

We demonstrated the benefits of the application of physical simulation for edge computing algorithms and distributed systems, especially in industrial data stream systems. For evaluation, an exemplary HiL setup was build for the VICO of an overall system consisting of an algorithm of the middleware layer, i.e., an agent system for resource allocation in IIoT, and edge computing algorithms of the logic layer.

The experiments prove the suitability of the proposed approach, i.e., the coupling to a traditional OT simulation environment for the validation of new edge computing algorithms for industry, according to Table 4.

Since this article is intended to draw attention to the wide range of potential applications of simulation for edge computing in the IIoT, the experiments performed do not claim to be exhaustive. Rather, they are intended to demonstrate benefits and feasibility in general by means of exemplary use cases.

Two main aspects are of relevance with regard to the success of the deployment of our proposal. Firstly, the simulation tool must be suitable for the particular use case, i.e., not every feature of a simulation tool is relevant for every use case. Capabilities such as real-time simulation or visualization must be taken into account when selecting the tool. The possibility to force values in the simulation environment during run time allows to manually trigger faulty states. This functionality is required for, e.g., the evaluation of anomaly detection algorithms. The second aspect that must be emphasized is the dependence on the quality of the simulation model. As with any application of simulation, the quality of the results depends directly on the quality of the simulation model.

When considering the proposed method for the described use cases in Section 5, the following limitations must be taken into account:The presented approach focuses on the industrial use, as it is to be expected that suitable simulation models are available or can easily be created with the existing tools. Additional application might be possible in the context of vehicles and vehicle fleets. A transfer to the IoT in general might be reasonable and possible but has to be examined for the respective use case, since physical simulation is not available for all use cases;The success of the simulation coupling highly depends on the quality of the simulation model, the suitability of the tool, and the infrastructure;The available functions of the simulation tools can be a limiting factor. Since the use case of HiL and SiL originates from the VICO of control applications on control units, some simulation tools are partly not yet able to run other programming languages than IEC 61131-3 code or to connect to other devices than PLCs.

## 8. Conclusions

In this article, we propose the further usage of simulation of industrial components or plants and present different approaches to combine this type of simulation with edge computing. Originating from SiL and HiL for control units, simulation can find further use in different stages of the life cycle of an industrial plant to enable and further advance edge computing in IIoT. After presentation of the method, including different possible architectures, an overview of the wide field of applications is given. It is divided into use cases for the VICO of distributed systems and edge computing in general. This is followed by an experimental evaluation. The experiments emphasize the proposal of this work.

For use in practice, the continuous use of simulation models is recommended. So far, data processing algorithms are usually trained and tested with their own environment. This can either be a recorded data set or a separate implemented environment. Due to the different development environments, it is difficult to test the algorithms in interaction before they are deployed at the production site. The continuous use of the simulation models (often already available) for these different kinds of edge computing algorithms enables comprehensive system testing and VICO in the final stages of development before deployment and commissioning in the real world. If this is ensured, the proposed use of simulation leads to many advantages such as time, cost, and resource savings, as well as technological advances and quality improvements that are hard to achieve without simulation. Since data-based business models are increasingly relevant in the I4.0, it is likely that edge computing and data processing will also become an additional part of the existing development process of industrial components and plants. Thus, this software application part is to be parallelized with the existing engineering and design process.

In a next step, especially for machine learning training, a comparison between training results in simulation and inference in the real system, as well as quantitative measurements about cost and time savings and quality benefits compared with the traditional procedures, are of interest to examine.

In the future, it might be interesting to combine the simulation approach with the described emulation framework in [65], which also has an interface for simulation as well as for physical hardware. Unfortunately, the available simulation interface is only suited for network simulators so far.

Additional advances in simulation tools, i.e., interfaces for additional edge device types or compatibility with programming languages deviating from the IEC 61131 standard, will take the use of simulation in the field of edge computing in I4.0 to the next level.

It is to be expected that the size and complexity of distributed system will continue to increase. Thus, edge computing applications and systems such as DSPSs will become more complex as well. This will result in a higher demand for VICO. The authors believe that VICO will be a key tool to help engineers in the development, implementation, and maintenance of these new-generation DSPSs.

## Figures and Tables

**Figure 1 sensors-23-03545-f001:**
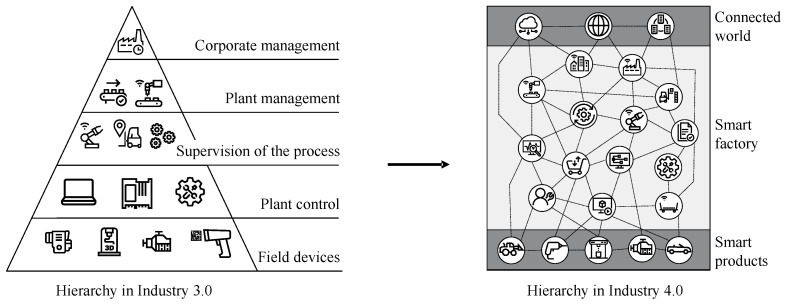
Hierarchy changes from Industry 3.0 to I 4.0 according to [19].

**Figure 2 sensors-23-03545-f002:**
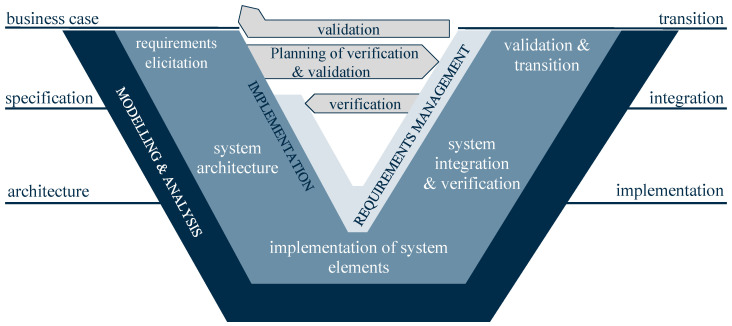
V-Model for development of mechatronic and CPSs [37].

**Figure 3 sensors-23-03545-f003:**
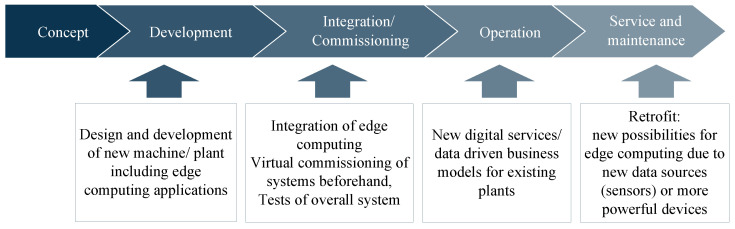
Application of simulation for VICO of DSPSs in different life cycle stages of a plant.

**Figure 4 sensors-23-03545-f004:**
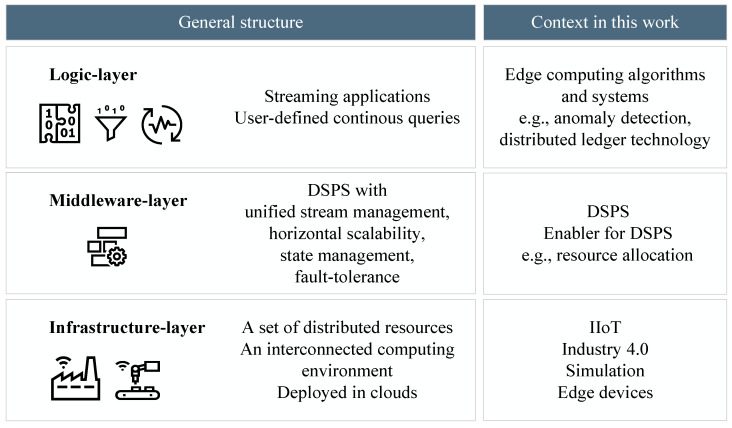
Three-layer architecture of a data stream system (according to [10]) in the context of this work.

**Figure 5 sensors-23-03545-f005:**
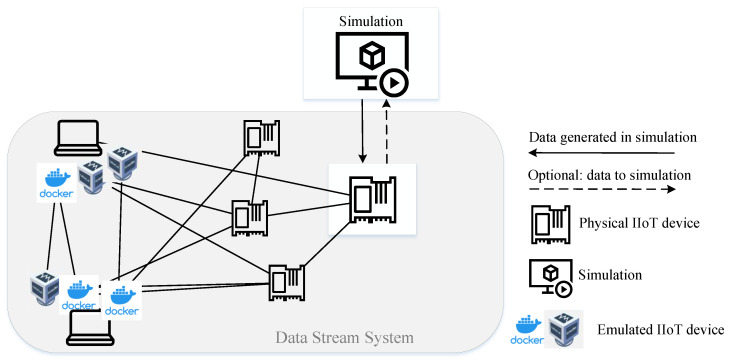
Architecture 1: one device coupled with a simulation and connected to a further network.

**Figure 6 sensors-23-03545-f006:**
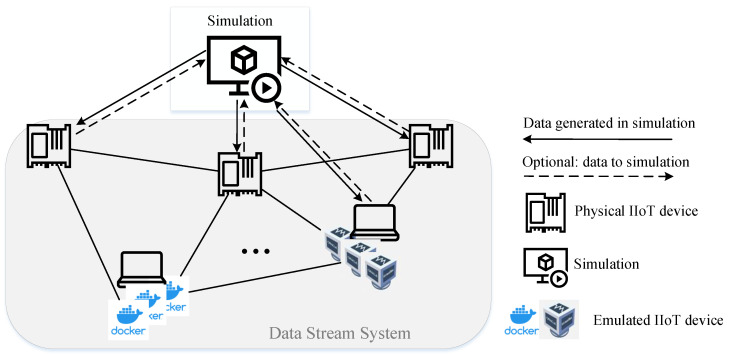
Architecture 2: several networked devices combined with one simulation.

**Figure 7 sensors-23-03545-f007:**
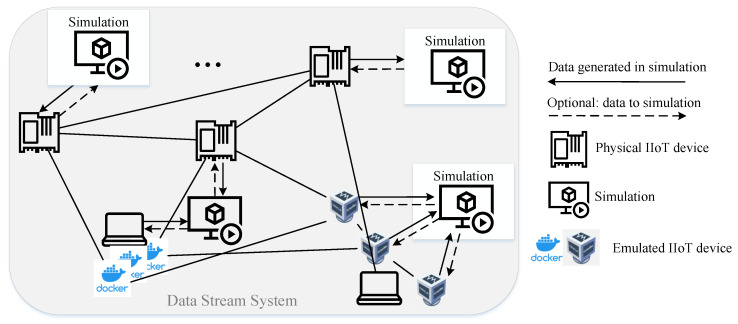
Architecture 3: several networked devices and distributed simulations.

**Figure 8 sensors-23-03545-f008:**
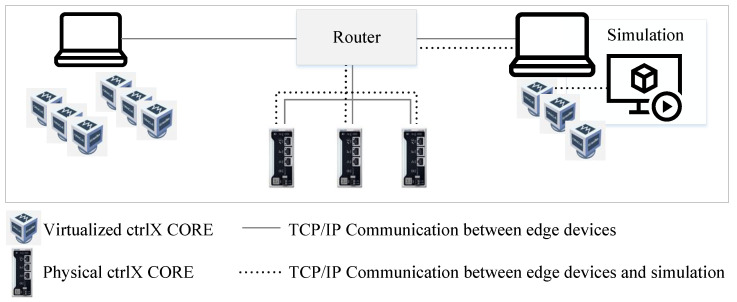
Experimental setup with a network of control units connected to a simulation model.

**Figure 9 sensors-23-03545-f009:**
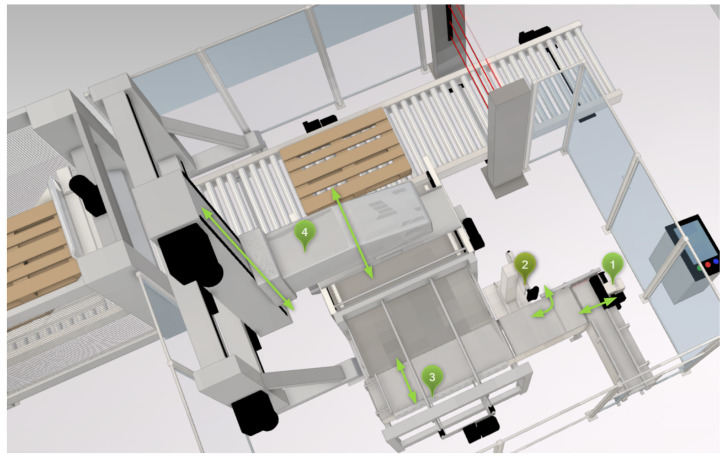
Simulation model used for evaluation: end-of-line palletizing portal packaging system. Motion controlled by PLC applications are marked in green. Physical control units: Nr. 1 pusher, Nr. 2 rotary vane, and Nr. 3 carriage. Virtualized control unit: Nr. 4 gripper arm.

**Figure 10 sensors-23-03545-f010:**
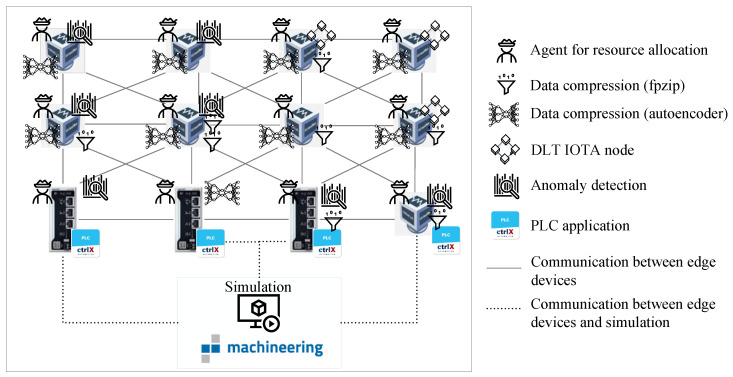
Illustration of the overall system in evaluation of the middleware layer.

**Figure 11 sensors-23-03545-f011:**
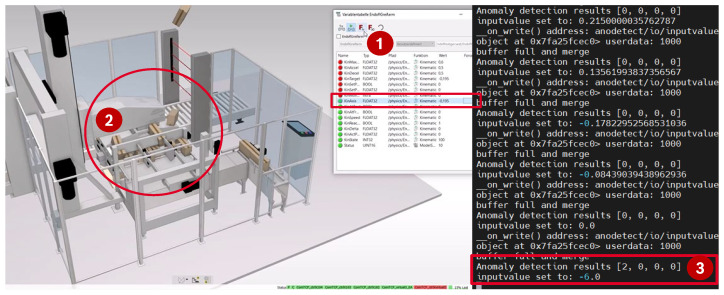
Manually initiated anomaly in the simulation environment and successful detection. 1. Manually forcing a faulty value (position gripper arm). 2. Visualization of abnormal behavior of the plant. 3. Successful detection by the anomaly detection algorithm on an edge device.

**Table 1 sensors-23-03545-t001:** Number of results for different search queries using Google Scholar (September 2022).

Results	Search Query
105	“edge computing” AND “simulation” AND “hardware-in-the-loop” AND IIoT
47	“edge computing” AND “virtual commissioning” AND “machine learning” AND “IIoT”
177	“distributed stream processing system” AND edge AND simulation AND IoT
6	“distributed stream processing system” AND edge AND simulation AND IIoT
48	“distributed stream processing system” AND “edge computing” AND simulation AND IoT
4	“distributed stream processing system” AND “edge computing” AND simulation AND IIoT
2	“distributed stream processing system” AND edge AND “process simulation”
0	“distributed stream processing system” AND edge AND “dynamic simulation”
0	“distributed stream processing system” AND edge AND “physical simulation”

**Table 3 sensors-23-03545-t003:** Considered data streams generated by the simulation for stream processing.

Stream Variable	Description	Data Type	Range
Selected streams of physical control unit 1 (192.168.8.102)			
Pusher_AXIS	Position of the pusher	float32	0 to 0.25
Selected streams of physical control unit 2 (192.168.8.103)			
Gleitschiene_Links_Axis	Position of the sliding rail (left)	float32	0 to 0.025
EndOfLaufwagen_ACT_SPD	Current speed of the carriage	float32	−0.8 to 0.8
EndOfLaufwagen_Axis	Position of the carriage	float32	0 to 1.5
EndOfLaufwagen_DLT_POS	Delta actual value–reference value of carriage position	float32	0 to 1.5
Selected streams of physical control unit 3 (192.168.8.104)			
Drehschieber_Achse_101_AXIS	Position of rotary vane	float32	0 to 0.13
Selected streams of virtual control unit 4 (192.168.8.83)			
EndoflHubgeruest01_ACT_SPD	Current speed of the portal	float32	max. 1.4
EndoflHubgeruest01_AXIS	Position of the portal	float32	−2 to 0
EndoflHubgeruest01_DLT_POS	Delta actual value–reference value portal position	float32	−2 to 0
EndoflGreifarm01_ACT_SPD	Current speed of the gripper arm	float32	max. 0.6
EndoflGreifarm01_AXIS	Position of the gripper arm	float32	−0.3 to 5
EndoflGreifarm01_DLT_POS	Delta actual value–reference value of gripper arm position	float32	−0.3 to 5

**Table 4 sensors-23-03545-t004:** Conducted experiments.

Experiment	Description	Success
Overall system		
Evaluation of overall system	All algorithms run together and the system interacts with a simulated plant	√
• Run PLC application	Controlling simulation model via real and virtualized PLCs	√
• Run MAS	Agent, watchdog, and resource monitoring as snaps on control unit	√
• Run IOTA	Faucet, masterpeer, and replica nodes as snaps on control unit	√
• Run anomaly detection	Anomaly detection algorithm *EEM-KDE* as snap on control unit	√
• Run data compression	Autoencoder and fpzip as snaps on control unit	√
Middleware-layer		
Resource allocation	Agents trigger task execution or forwarding depending on available resources	√
Load test	Watchdog frees resources and sends task back to task queue for reassignment	√
Adaptivity to dynamic changes	Adding/removing agents or data streams during runtime	√
Logic-layer		
Anomaly detection	Detection of manually forced anomalies	√
Data compression	Both autoencoder and fpzip compress and decompress data	√
IOTA		√
• Adaptivity to dynamic changes	Adding/removing nodes during runtime	√
• Tangle	Tangle is built and synchronized	√
• Smart Contracts	Execute SCs that interact with simulation	planned

## Data Availability

Not applicable.

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
