# Peer review of "Virtual Commissioning of Distributed Systems in the Industrial Internet of Things"

_sensors, 2023, doi:10.3390/s23073545_

Round 1
Reviewer 1 Report
In this paper, the authors have presented research on the Virtual Commissioning of distributed systems in Industry 4.0 (I4.0). This research topic is very interesting.
1) Besides contribution, and benefits, please mention the motivation of this research in the introduction section.
2) In related work it would be great if you mention the pros and cons in a tabular form.
3)The figure quality should be improved in the overall paper.
4)The simulation tool details should be mentioned in section 6.
5) Please modify Section 7 and add implications and your analyses in this section.
6) Conclusion is not mentioned in this study. Therefore please add at the end and also mention future work should be mentioned in this section.
Author Response
Dear Reviewer,
We thank you for the constructive feedback and hope to have satisfactorily revised all mentioned points of criticism.
Please see the attachment.
Kind regards,
Julia Rosenberger

Reviewer 2 Report
The proposal focuses on the further usage of simulation of industrial components or plants and presents different use cases and approaches to combine this type of simulation with edge computing. The proposal presents an interesting topic; however, the following aspects were identified:
1. A section of related works divided into three sub-sections is presented, however, the description of the main differences of the related works with respect to the proposal of this work that allows visualizing its importance and above all its novelty is not presented.
2. In section 3. "State Of The Art" Table 1 is presented which presents the GoogleScholar queries, however, there is no reference to this table in this section, perhaps the table is out of place.
3. Figure 2 can be omitted and better to present in writing the context of the dynamic changes in the IioT. The same is the case with figure 3, it is better just to describe the context of the figure. Likewise in figure 4 it is preferable to summarize in a few lines the evolution of the DSPSs.
4. Consider restructuring and shortening section 2. Background, as it is very long and tends to confuse.
5. Check the format of the citation [39], the point is at the end.
6. It is suggested that an image or table should not appear before or after the title or subtitle of any section or subsection. This happens when presenting figures 2, 4, 5, 7, 10 and 18.
7. It is necessary to include a section of conclusions where the authors present and describe their own conclusions, as well as to indicate the future work after the work done.
Author Response

(The authors gave the same response as above.)

Round 2
Reviewer 1 Report
The authors have nicely incorporated all necessary changes in the manuscript. Thanks
Reviewer 2 Report
The authors correctly followed the suggestions of the review.